# Ethylene Promotes Expression of the Appressorium- and Pathogenicity-Related Genes via GPCR- and MAPK-Dependent Manners in *Colletotrichum gloeosporioides*

**DOI:** 10.3390/jof8060570

**Published:** 2022-05-26

**Authors:** Dandan Ren, Tan Wang, Ganghan Zhou, Weiheng Ren, Xiaomin Duan, Lin Gao, Jiaxu Chen, Ling Xu, Pinkuan Zhu

**Affiliations:** School of Life Sciences, East China Normal University, Shanghai 200241, China; 52201300003@stu.ecnu.edu.cn (D.R.); 20091036@nynu.edu.cn (T.W.); 51191300004@stu.ecnu.edu.cn (G.Z.); 52191300004@stu.ecnu.edu.cn (W.R.); 15216625967@163.com (X.D.); gaolin199801@163.com (L.G.); jxchen6854@163.com (J.C.)

**Keywords:** ethylene, anthracnose, appressorium, postharvest disease, MAPK, GPCRs

## Abstract

Ethylene (ET) represents a signal that can be sensed by plant pathogenic fungi to accelerate their spore germination and subsequent infection. However, the molecular mechanisms of responses to ET in fungi remain largely unclear. In this study, *Colletotrichum gloeosporioides* was investigated via transcriptomic analysis to reveal the genes that account for the ET-regulated fungal development and virulence. The results showed that ET promoted genes encoding for fungal melanin biosynthesis enzymes, extracellular hydrolases, and appressorium-associated structure proteins at 4 h after treatment. When the germination lasted until 24 h, ET induced multiple appressoria from every single spore, but downregulated most of the genes. Loss of selected ET responsive genes encoding for scytalone dehydratase (*CgSCD1*) and cerato-platanin virulence protein (*CgCP1*) were unable to alter ET sensitivity of *C. gloeosporioides* in vitro but attenuated the influence of ET on pathogenicity. Knockout of the G-protein-coupled receptors CgGPCR3-1/2 and the MAPK signaling pathway components CgMK1 and CgSte11 resulted in reduced ET sensitivity. Taken together, this study in *C. gloeosporioides* reports that ET can cause transcription changes in a large set of genes, which are mainly responsible for appressorium development and virulence expression, and these processes are dependent on the GPCR and MAPK pathways.

## 1. Introduction

Fresh fruits are valuable dietary sources for human health; however, they are also attractive targets for phytopathogenic fungi, leading to fruit decay and economic losses. As the fruits ripen and senescence, the physicochemical properties of the host tissues become increasingly susceptible to microorganism infections. On the other side, the pathogens should be capable of recognizing fruit ripening status to orchestrate their infection activities, although the mechanisms by which they do this remain unclear.

The gaseous phytohormone ethylene (ET) is involved in many aspects of plant growth and development, including senescence, organ abscission, and fruit ripening [1,2,3]. In plants, ET is perceived by transmembrane-receptor proteins localized in the endoplasmic reticulum (ER) belonging to the ET receptor family that harbors homology to bacterial two-component histidine kinases [4,5]. The ET signal transduction pathway in plants ultimately leads to regulating the expression of ET-dependent genes [6,7]. In flesh fruits, ET regulates ripening by coordinating the expression of genes that participate in multiple biological processes, such as the increase in respiration, autocatalytic ET production, pigment accumulation, change in texture, and the buildup of overall fruit quality traits [8]. As such, ET is known as the fruit-ripening hormone. Because of its gaseous nature, ET can diffuse rapidly throughout plant materials, cross cell membranes, and escape the plant as an environmental signal, which could play important roles in interactions between host plants and pathogenic microbes [9].

In contrast to healthy fruits, the diseased ones usually release notably increased ET, which can thereby stimulate the ripening and rotting of nearby fruits, fulfilling the probable capacity of ET to accelerate fruit spoilage in holistic scales [10,11]. ET is also involved in a variety of plant defense responses to microbial attacks [12,13]. A large number of studies have shown that, depending on the nature of the plant–microbe interaction, ET may reduce, enhance, or have no effect on disease development [14]. Most studies have focused on the roles of ET signaling in the immunity of host plants, leading to the commonly accepted view that ET and jasmonic acid signaling components are involved in resistance against necrotrophic pathogens [15,16]. However, the effects of ET on the pathogen side are relatively less understood.

It is noteworthy that plant pathogenic fungi are capable of sensing ET. For example, ET can promote the spore germination and appressorium formation of *Colletotrichum* spp. [17]; In contrast, 100 ppm and 1000 ppm exogenous ET released by ethephon mildly inhibited the spore germination of *Botrytis* spp., while low concentrations of ET showed a negligible effect [18]. Treatment of *B. cinerea* with 2,5-norbornadiene, a kind of ET receptor competition inhibitor, can inhibit the germination of spores, while ET can eliminate the inhibitory effect [19]. Similar phenomena have been found in *Alternaria alternata* [20]. Additionally, the expression of the pathogenic gene *bcspl1* of *B. cinerea* could be induced by ET [21]. In *Aspergillus flavus*, ET can suppress aflatoxin production, and inhibit the expression of aflatoxin-synthetic genes (*aflR* and *aflD*) [20,22]. However, the effect of ET on the overall gene expression profiles of plant pathogenic fungi has not been reported yet, leaving a knowledge gap in our understanding of the mechanisms underlying the effects of ET on fungal growth, development, and pathogenicity.

*Colletotrichum gloeosporioides* is one of the most common pathogens causing postharvest diseases in varied fruits and vegetables [23,24,25]. *C. gloeosporioides* is a characteristically hemibiotrophic fungus that can survive in the plant cuticle or epidermal cells until the fruits ripen [26]. In other words, *C. gloeosporioides* prefers to remain under the latent infection stage in immature fruits and vegetables, with the anthracnose symptoms occurring as the host matures. The conidia of *C. gloeosporioides* can form a typical dome-shaped infection structure called appressorium [27]. The fungal spores need to spend their substance storage inside to provide energy for the formation of appressoria, and then rely on the newly developed cell structures to infect plants and obtain nutrients to maintain the growth and reproduction [28,29]. *C. gloeosporioides* represents a typical plant pathogenic fungus showing obvious ET-sensitive responses, which include spore germination and appressorium formation [17]. Therefore, ET may be recognized as a signal of fruit ripening by this pathogenic fungus to initiate the occurrence and development of anthracnose disease.

The G-protein-coupled receptors (GPCRs) are the largest class of cell surface receptors in eukaryotes, sensing sense environmental cues to initiate intracellular G-protein signaling that coordinates biological responses [30,31]. In *Magnaporthe grisea*, the G protein subunits, MagB and Mgb1, are required in a cAMP-dependent manner for recognition of host surface hydrophobicity during appressorium formation [32,33]. However, cAMP does not rescue defects in appressorial penetration and infectious growth in *Mgb1* and *Mgg1* mutation strains [33,34]. In addition, the putative GPCR Pth11 also regulates appressorium formation, presumably through sensing surface hydrophobicity and plant cutin monomers [35]. Thus, Pth11/MagB/Mgb1/Mgg1 could comprise a GPCR/G protein system that controls appressorium differentiation in response to molecular cues from the host plant. Interestingly, a Gα null mutant (Δ*bcg1*) in *B. cinerea* was ET insensitive [21], and one identified GPCR (BcGPR3) demonstrated the ability to bind ET [36]. Cumulatively, these studies imply that certain GPCR/G protein systems could be involved in ET sensing in fungal pathogens. 

Upon perceiving exogenous stimuli by the GPCR/G protein system, the signals could be further relayed to intracellular cyclic adenosine monophosphate (cAMP) or mitogen-activated protein kinase (MAPK) signaling pathways, vital regulators for the establishment of appressorial structure and function in plant pathogenic fungi [37,38,39]. The studies with *C. gloeosporioides* revealed that deletion of the gene encoding the catalytic subunit of cAMP-dependent protein kinase A retarded the formation of appressorium [39]. Meanwhile, *C. gloeosporioides* containing mutated mitogen-activated protein kinase (CgMK1) could not form appressorium. Exogenous addition of cAMP failed to restore the mutant’s defect, indicating that CgMK1 may function downstream or independent of cAMP-mediated signaling for appressorium formation [40]. Thus, whether the intracellular secondary signaling pathways are involved in ET responses to regulate appressorium development is worth investigating.

The present study aimed to uncover the molecular events regulated by ET to affect the development and pathogenicity of *C. gloeosporioides*. Through transcriptome analysis, the ET-regulated genes related to appressorium formation and pathogenicity of *C. gloeosporioides* were systematically explored. In addition, reverse genetic study determined the functions of specific ET signaling and responsive genes, leading to the speculation that ET may promote appressorium formation and pathogenicity of *C. gloeosporioides* through the GPCRs and MAPK signaling pathways.

## 2. Materials and Methods

### 2.1. Fungal Pathogen and Host Plant Materials

Single-spore strain EX2016-02 of *C. gloeosporioides* has been isolated and characterized in a previous study [41], and this strain was used as a recipient for genetic modification. All the fungal strains used in this study are listed in Table 1. To generate fungal conidial suspension, approximately two-week-old, sporulating cultures on potato dextrose agar (PDA) plates were flooded with sterile water. The resultant conidial suspension was measured for concentration and adjusted to 3 × 10^5^ conidia mL^−1^ using a hemocytometer. *Arabidopsis thaliana* (Col-0) was grown in vermiculite soil at 23 °C under short-day conditions (8 h light/16 h dark) for 4 weeks to obtain detached leaves for the inoculation assay. 

### 2.2. Ethylene Treatment and Sample Preparation

Gaseous ethylene (ET) was purchased from Pujiang Special Gas Co., Ltd., Shanghai, China. To examine the effects of ET, the samples were placed in humidified plastic boxes with tight sealing caps. Pure ET was injected into the boxes to achieve specified concentrations, which were confirmed via gas chromatographic analysis according to the reported method [43]. Meanwhile, the samples placed in the boxes without ET input were used as controls. 

To harvest fungal samples for transcriptome study, conidia suspension was poured into 90 mm diameter glass Petri dishes, with 20 mL per dish. The Petri dishes were placed in sealed plastic boxes. Control and ET (200 ppm) treatment were performed as shown above. After incubation for 4 or 24 h, the supernatant liquid was gently poured out, and the remaining conidia and germlings were scratched from the bottom of the dishes by using a cell scraper. Three independent samples were collected for each treatment, and the experiment was repeated three times. The resulted samples were immediately frozen in liquid nitrogen, followed by thorough lyophilization before RNA extraction.

### 2.3. Total RNA Extraction

Total RNA was extracted using QIAGEN Reagent (Hilden, Germany) and then treated with DNase I according to the manufacturer’s instructions. The quality of extracted total RNA was verified using a Nanodrop 2000 (Thermo Fisher Scientific Inc., Waltham, MA, USA) and Agilent Bioanalyzer 2100 system (Agilent Technologies, Santa Clara, CA, USA). The total amount of RNA required for a single sample of transcriptome test was not less than 5 μg, and the optical density value of 260 over 280 was between 1.8 and 2.2. 

### 2.4. Library Preparation and RNA Sequencing

The RNA library was constructed using the Illumina Truseq^TM^ RNA sample prep kit method (Origingene biotechnology company, Shanghai, China). Briefly, three independent samples were collected for each treatment. Total RNA was extracted and enriched by oligo-dT selection. The mRNA samples were fragmented and reverse-transcribed to double-strand cDNA using the random primers and reverse transcriptase. The sticky ends of synthesized cDNA were repaired to blunt ends using the End Repair Mix, and then to the 3’ end, an A base was added to connect to the Y-shaped adaptor. Paired-end reads of 150 bases were obtained using Illumina HiSeq 2000 and converted into sequence data through Base Calling.

### 2.5. Transcriptomic Data Analysis

The clean reads were filtered using cutadapt (v1.16) from the low-quality reads, which were stored in FASTQ format [44]. Genome assemblies of *C. gloeosporioides* Cg-14 (NCBI accession number SUB133583) were used as a reference sequence, and the reads were aligned to the reference sequence with hisat2 software (v2.1.0) [45]. Fragments per kilobase per million (FPKM) for differential expression using StringTie software (v1.3.3b) were calculated [46]. Differentially expressed genes (DEGs) (fold change ≥ 2, adjusted *p*-value ≤ 0.05) were identified with the DEseq2 algorithm [47]. Heat map visualization was performed using ggplot2 and pheatmap packages, and the overall number of differential genes were counted using the UpSetR package [48,49]. Principal component analysis (PCA) was calculated from the expression data using ggpubr, ggthemes, and gmodels packages. Blast2GO software was used for functional annotation of the transcripts [50]. GO analysis and visualization were performed using the R language GOcircle package.

### 2.6. Quantitative Real-Time Reverse Transcription PCR (qRT-PCR) Analysis

For qRT-PCR analysis, 1 μg of each RNA sample was used for reverse transcription with the Prime Script™ RT reagent kit (Perfect Real Time) (TakaRa Biotechnology, Co., Dalian, China). The real-time PCR amplifications were conducted in a CFX96TM Real-Time System (BIO-BAD, Inc., California, USA) using TakaRa SYBR Premix Ex Taq (TakaRa Biotechnology, Co., Dalian, China). For each sample, the expression of the *Actin* gene was used as an internal reference. The primers for the qRT-PCR assays are listed in Appendix A. There were three replicates for each sample, and the experiment was repeated three times. Relative expression levels of the tested genes were calculated using the 2^−∆∆Ct^ method [51].

### 2.7. Fungal Transformation

Molecular manipulation and PEG-mediated protoplast transformation as described in a previous study were used for the purposes of deleting target genes in *C. gloeosporioides* [42]. For the purposes of expressing exogenous genes, *Agrobacterium tumefaciens*-mediated transformation (ATMT) was used. The recombinant plasmids based on the backbone pAg1-H3 vector were constructed and transformed into *A. tumefaciens* strain Agl1 for ATMT according to the reported method [52].

### 2.8. Fluorescence-Based Reporter Assay

The fluorescence-based reporter assay was conducted to compare the degree of ET responses in *C. gloeosporioides*. Briefly, the promoters of relevant ET-responsive genes were fused with *GFP* and assembled into the pAg1-H3 vector at the SacI digestion site via a one-step cloning kit (Yeasen, Shanghai, China). The resulted recombinant plasmids were transformed into the fungus by the ATMT method. The conidial suspension (3 × 10^5^ conidia mL^−1^) of the transformants was dropped on sterilized glass slides and incubated in humidified plastic boxes for 24 h. Control and ET treatments were conducted as described above. Observation of the fluorescent expression of transformants with a fluorescence microscope (Axio Imager 2, ZEISS, Jena, Germany). For each treatment, three replicates of 200 conidia were randomly selected for observation recording.

### 2.9. Appressorial Formation and Pathogenicity Assay

To examine the effects of ET on conidial germination and subsequent hyphal development, a conidial suspension (3 × 10^5^ conidia mL^−1^) was dropped on sterilized glass slides and incubated in humidified plastic boxes. Control and ET treatments were conducted as described above. After incubation for 4 or 24 h, conidial germination and appressorium formation rates were recorded. For each treatment, three replicates of 300 conidia were randomly selected for statistical analysis.

In the pathogenicity assay, *A. thaliana* leaves were detached and inoculated with 6 µL conidia suspension. The inoculated materials were kept in humidified boxes containing either 0 or 200 ppm ET at the beginning of the assay. The boxes were placed under light and incubated at 25 °C. After 24 h, the samples were subjected to DAB staining analysis.

### 2.10. Staining Analysis for Reactive Oxygen Species in Plant Leaves

Reactive oxygen species (ROS) accumulation upon pathogen infection in plant leaves were visualized by DAB (3,3′-diaminobenzidine) staining. Briefly, 220 mg DAB was added to 200 mL sterile water. The pH of the solution was adjusted to 3.7 to completely dissolve the contents. The solution was mixed with 110 mL Tween 20 (0.05% *v*/*v*) and 0.8 g Na_2_HPO_4_ to obtain the DAB dye solution. The samples for analysis were stained in the freshly prepared solution by shaking at 100 rpm in the dark for 4 h. After staining, the samples were boiled in 95% ethanol to decolorize. Photographs of the samples were analyzed with ImageJ software to calculate the depth of staining according to the reported method [53].

### 2.11. FDA-PI Dual Fluorescence Staining

FDA (fluorescein diacetate) was dissolved in acetone to obtain a final concentration of 30 mg mL^−1^, and PI (propidium iodide) was dissolved in DPBS buffer (Dulbecco’s phosphate-buffered saline) to obtain a final concentration of 10 mg mL^−1^. Five microliters of the above solutions were diluted to 1 mL with sterile water to obtain a fresh staining solution. The conidial suspension was placed on the glass slide to germinate for 4 h or 24 h, and the staining solution was added to the samples. After staining at room temperature for 30 min, the dyes were drained, and the samples were washed 3 times with sterile water. Images were acquired with the fluorescence microscopic analysis as follows: live cells were stained with FDA to show a bright green fluorescent signal (530 nm) upon excitation light (488 nm), while in contrast, dead cells were stained with PI to emit a red fluorescent signal (617 nm) under green excitation light (535 nm).

### 2.12. Calcofluor White-EosinY Double Fluorescence Staining 

The double staining assay was conducted according to the previous method [54], in which chitin and its deacetylated form chitosan were stained by Calcofluor white (18909-100ML-F, Sigma-Aldrich, St. Louis, Missouri, USA) and EosinY (A600190-0025, Sangon Biotech, Shanghai, China), respectively. The conidial suspension was placed on the glass slide for germination. After staining in the dark at room temperature for 10 min, the staining solution was removed, and the samples were analyzed with a fluorescent microscope. Fluorescent signals of calcofluor white and Eosin Y were captured using 405 and 561 nm excitation wavelengths, respectively.

### 2.13. Statistical Analysis

Statistical analyses were carried out with the GraphPad Prism 8.0 software. Bar charts represent mean values with standard deviations; differences were considered significant at *p* < 0.05 (*).

## 3. Results

### 3.1. Ethylene Accelerated Conidial Germination, Appressorium Formation, and Pathogenicity of C. gloeosporioides

The conidia of *C. gloeosporioides* treated with ET demonstrated an appressorium formation rate of 71.96% on glass slides at 4 h post-inoculation (hpi), in contrast to that of the control (CK) value of 10.64%. At 24 hpi, the gap between the CK (86.08%) and ET treatment (91.39%) groups in appressorium formation rates was not comparable as it had been at 4 hpi; however, the appressorium yield per single conidium was higher in the ET group than that of CK group at 24 hpi (Figure 1A,B). FDA and PI double staining showed that germinated spores in the CK- and ET-treated groups survived at 4 hpi, as these cells were filled with FDA staining signal (bright green). However, at 24 hpi, the proportion of PI-stained cell areas was higher in the ET-treated samples, in other words, the prolonged ET treatment resulted in stimulating appressorium development, which was accompanied by apoptotic processes (Appendix A). Fluorescent microscopic analysis with the *Cg-gfp* strain demonstrated that ET could also promote the appressorium formation on the *Arabidopsis* leaf surface. Additionally, the accumulation of reactive oxygen species (ROS) at the inoculation site as visualized by DAB staining showed that there were stronger ROS in the ET group than in the CK (Figure 1C). These results indicate that ET can significantly promote the appressorium differentiation and host infection of *C. gloeosporioides*.

### 3.2. ET Treatment Altered Transcript Levels of a Large Set of Genes in Germinating Conidia of C. gloeosporioides

In order to reveal the molecular events related to the effects of ET on *C. gloeosporioides*, RNA-seq assays were conducted with the germinating conidia at 4 and 24 hpi. Compared with the CK group, ET treatment resulted in 833 upregulated and 997 downregulated genes at 4 hpi. However, ET treatment only upregulated 78 genes but downregulated 2649 genes at 24 hpi (Figure 2A). The up- and downregulated genes by ET treatment at both 4 hpi and 24 hpi were 9 and 518, respectively (Figure 2B). PCA revealed that there were significant differences in gene expression between the CK-4hpi VS CK-24hpi, ET-4hpi VS CK-4hpi, and ET-24hpi VS CK-24hpi groups. However, the ET-4hpi and CK-24hpi groups were overlapped in PCA (Appendix A, Figure 2C). The gene expression values obtained from RNA-seq were validated using the qRT-PCR assay. The expression levels of 27 randomly selected *C. gloeosporioides* genes demonstrated a strong correlation (R^2^ = 0.9089) between the results obtained with the two techniques (Figure 2D, Appendix A).

### 3.3. Functional Enrichment Analysis of Differentially Expressed Genes

The differentially expressed genes (DEGs) as influenced by ET treatment were subjected to the GO (gene ontology) classification analysis. The results showed that anchored component of membrane (GO:0031225), extracellular region (GO:0005576), cell surface (GO:0009986), intrinsic component of plasma membrane (GO:0031226), fungal-type cell wall (GO:0009277), and other cellular-component-related genes were significantly upregulated by ET at 4 hpi. Meanwhile, the expression of a large number of genes related to biological processes such as response to stress (GO:0006950), response to abiotic stimulus (GO:0009628), response to oxidative stress (GO:0006979), and DNA synthesis involved in DNA repair (GO:0000731) were downregulated. At 24 hpi, the DEGs downregulated by ET were enriched in the reproduction (GO:0000003), cytokinesis (GO:0000910), transcription (GO:0006355), cellular protein modification process (GO:0006464), and phosphate-containing compound metabolic process (GO:0006796) (Figure 3, Appendix A).

As the “extracellular region” GO enrichment included the largest number of upregulated DEGs at 4 hpi, it was further found that these DEGs mainly encode two types of products: extracellular hydrolases and extracellular structural components. As shown in Appendix A, ET treatment can significantly promote the expression of the genes encoding for polysaccharide deacetylase, cutinase, multicopper oxidase, and cerato-platanin protein. At the same time, ET could also promote transcript levels of the genes encoding for cell wall proteins, appressoria structure proteins, glycolipid-anchored surface protein, and other structural proteins.

### 3.4. Hydrophobic Surface Binding Protein A and Cutinase-Related Genes Were Induced by ET Treatment at the Early Germinating Stage

Twelve most strongly induced DEGs by ET at 4 hpi are described in Table 2. Via Protein Blast analysis to characterize the conserved domains encoded by these DEGs, three genes (CGLO_03844, CGLO_00547, CGLO_13252) encoding for hydrophobic surface binding protein A domains (HsbA) were identified. Via the GFP-based reporter analysis, it was shown that ET treatment promoted the GFP-expressing signal when this reporter gene was driven by the promoters of *HsbA* genes (CGLO_03844 and CGLO_13252), suggesting that these *HsbA* genes are indeed highly sensitive to ET stimulus (Figure 4A,C, Appendix A). In *Aspergillus oryzae*, HsbA is adsorbed to the hydrophobic surface to recruit cutinase CutL1, which can degrade the hydrophobic cutin on the plant surface [55]. Interestingly, the genes encoding for cutinase transcription factor genes (CTF1α CGLO_17337, CTF1β CGLO_00958) and cutinase genes (CGLO_17126, CGLO_16577) of *C. gloeosporioides* were also significantly upregulated upon ET treatment (Figure 4B, Appendix A). 

### 3.5. ET Promoted the Melanin Synthesis and Chitin Deacetylation at the Appressorial Cell Wall

Appressoria of *C. gloeosporioides* are induced by ET with a characteristic accumulation of a dark pigment, the fungal DHN (1,8-dihydroxynaphthalene) melanin. The biosynthesis process of DHN melanin involves a series of key enzymes including polyketide synthase (PKS), 1,3,6,8-tetrahydroxynaphthalene (T4HN) reductase (T4HR), 1,3,8-trihydroxynaphthalene (T3HN) reductase (THR), scytalone dehydratase (SCD), and laccases [54,56]. Additionally, the fungal transcription factors, CMR1 and its orthologs, are known to regulate the expression of the DHN melanin biosynthesis genes in fungi [57]. We found that the genes encoding for melanin synthases of *C. gloeosporioides*, including *CgPKS1* (CGLO_05047), *CgT4HR1* (CGLO_10812), *CgTHR1* (CGLO_00442), and *CgSCD1* (CGLO_09685), as well as the transcription factor *CgCMR1* (CGLO_10813), were significantly upregulated upon ET treatment. Moreover, ET also stimulated the transcript levels of a group of laccase genes (CGLO_06458, CGLO_02601, CGLO_06095, CGLO_12977, CGLO_02394, and CGLO_04100), which could be responsible for polymerizing 1,8-DHN monomers into the final melanin products (Figure 5A, Appendix A).

Filamentous fungal cell walls are generally composed of chitin polysaccharides. However, this cell wall component can be modified by the fungi themselves for certain purposes, such as evading the host immune recognition and thus enhancing the pathogenicity of the fungi. Chitin deacetylases (CDA) are enzymes that catalyze the transformation of chitin into chitosan. The genome of *C. gloeosporioides* contains seven putative CDA-encoding genes, among which five CDA genes (CGLO_07057, CGLO_16772, CGLO_00212, CGLO_16754, and CGLO_02140) were upregulated by ET. Double-staining assay with Calcofluor and Eosin Y revealed that as appressorium formation proceeded, chitin was deacetylated to form chitosan at the appressorial zones (Figure 5B, Appendix A). These results indicate that the appressorial cell wall of *C. gloeosporioides* is characterized by accumulating melanin and chitosan, which can be highly promoted by the presence of ET.

### 3.6. Pathogenicity Enhancement of C. gloeosporioides in Response to ET Was Partially Dependent on the Upregulation of Melanin Synthase and Effector Genes

Hitherto, the study confirms that ET can promote appressorium formation and infection of *C. gloeosporioides*. Accompanying ET-induced genes were further screened for their contribution to the ET-regulated pathogenicity. Three selected ET-responsive genes were targeted for knockout: *CgCAP22* (CGLO_01483), predicted to encode the homolog of appressorium-specific protein CAP22 [58]; *CgCP1* (CGLO_12973), encoding the virulent effector cerato-platanin [59]; and *CgSCD1* (CGLO_09685), encoding the melanin biosynthesis enzyme [42]. The resultant mutant strains of Δ*Cgcap22*, Δ*Cgscd1*, and Δ*Cgcp1* showed no significant difference in the growth rate compared to the WT (Figure 6A).

Furthermore, in vitro assays demonstrated that ET induced appressorium development in all the tested mutants by comparable level in the WT, suggesting that the mutants retain sensitivity to ET (Figure 6B). In contrast, in the infection assay on *Arabidopsis* leaves, ET treatment enhanced ROS accumulation at the inoculation sites of the WT and Δ*Cgcap22* mutant spores, while the Δ*Cgscd1* and Δ*Cgcp1* inoculation sites resulted in negligible ROS accumulation, irrespective of ET treatment or not (Figure 6C). Consequently, upregulating the melanin synthesis and the effector genes is likely of vital importance for ET to promote the pathogenicity of *C. gloeosporioides*.

### 3.7. ET Promoting Appressorium Formation and Virulence Was Dependent on MAPK and GPCR Pathways

The mitogen-activated protein kinases (MAPK) and G-protein-coupled receptor (GPCR) signaling pathways are known to be involved in sensing environmental signals and regulating spore germination in fungi. In this study, the following genes *CgMK1* (CGLO_17118) encoding for MAPK, *CgSte11* (CGLO_15819) encoding for MAPKKK, and *CgGPCR3-1* (CGLO_14522) and *CgGPCR3-2* (CGLO_16780) encoding for two GPCRs, were cloned and identified from the genome of *C. gloeosporioides*. Knock-out mutation via protoplast transformation and homologous recombination resulted in the corresponding mutants of Δ*Cgmk1*, Δ*Cgste11*, Δ*Cggpcr3-1*, Δ*Cggpcr3-2*, and Δ*Cggpcr3-1/2* double mutants. The growth rate of Δ*Cgmk1* and Δ*Cgste11* were significantly impaired in comparison with the WT strain. However, the Δ*Cggpcr3-1*, Δ*Cggpcr3-2*, and Δ*Cggpcr3-1/2* mutants demonstrated comparable growth rates as the WT (Figure 7A,C).

Concerning the ET-responsive phenotypic characters, the Δ*Cgmk1* and Δ*Cgste11* mutants showed defects in appressorium formation, but spore germination was slightly promoted by ET (Figure 7D). Moreover, the Δ*Cggpcr3-1*, Δ*Cggpcr3-2*, and Δ*Cggpcr3-1/2* showed elevated spore germination rates in response to ET; however, induction of appressorium formation by ET was attenuated in these GPCR mutants, among which the double mutant Δ*Cggpcr3-1/2* was the most extremely depressed (Figure 7B,D,E).

DAB staining further demonstrated that upon ET treatment, the inoculation sites with WT and Δ*Cggpcr3-1/2* showed stronger ROS accumulation than their corresponding control samples. In contrast, the Δ*Cgste11* inoculation sites resulted in negligible ROS accumulation in both control and ET treatment conditions (Figure 7F). Collectively, these results imply that the MAPK- and GPCR-mediated signaling pathways are likely involved in the ET-accelerated appressorium development; however, it is the MAPK pathway component that is required for mediating ET to promote the pathogenicity of *C. gloeosporioides* in the plant.

Moreover, gene expression analysis indicated that several ET-induced genes, including CGLO_09685 (*CgSCD1*), CGLO_12793 (*CgCP1*), CGLO_16754 (encoding for chitin deacetylase), CGLO_03844 and CGLO_13252 (encoding for hydrophobic surface binding protein A), and CGLO_15181 (hypothetical protein), were extensively impaired in ET responsiveness in the Δ*Cgmk1* and Δ*Cgste11* mutants. As for the Δ*Cggpcr3-1/2* double mutant, expression levels of the above genes except for CGLO_12793 (*CgCP1*) were also not sensitive to ET stimulus (Figure 8).

## 4. Discussion

Ethylene (ET) is the ripening hormone for fruit crops to achieve full maturation and attractive qualities. Microbial contamination and rotting are favored by fruit ripening, and usually lead to the release of more ET, thereby stimulating the ripening and rotting of nearby fruits. In this context, the influence of ET on both fruits and their rotting agent fungi should be precisely understood to achieve better management of postharvest losses. The ET signaling mechanisms in plants have been extensively elucidated since the 1990s, mostly due to a series of milestone findings based on elegant genetic analysis with the model plant *Arabidopsis* [60,61], while the ET action mode and mechanisms on plant pathogens are still largely unknown. The current study used transcriptomic and reverse genetic analysis to reveal the molecular response mechanisms to ET in the anthracnose fungus *C. gloeosporioides*.

*C. gloeosporioides* is a typical hemibiotrophic fungus that infects plants by forming appressoria. Spore germination and subsequent appressorium development of this fungus can be accelerated by ET [17]. Using the transgenic strain of *Cg-gfp* expressing GFP, this study confirms that ET can significantly promote the formation of appressoria in vitro and on *Arabidopsis* leaves. Fungal ET responses such as accelerated spore germination appear similar to the postulated ancestral phenomenon of ET motivating cell elongation in plants [62,63]. Moreover, appressoria are known as specialized cells or adhesion structures from which a penetration peg emerges to pierce the host tissue [64]. We found that ET enhanced appressoria development and ROS accumulation at the inoculation site on leaves, suggesting that ET could influence the outcome of the pathogen–host interaction by affecting the fungal side. These ET-sensing responses in fungi, exemplified by *C. gloeosporioides*, are thought to provide the pathogen an adaptive advantage by allowing them to accurately time their infections to the host’s ripening stages [17].

In order to reveal the molecular mechanisms underlying the ET responsive phenotypes of *C. gloeosporioides*, the transcriptome data of the germinating spores were analyzed. ET was found to cause a large number of upregulated genes at 4 hpi, with the genes coding for “extracellular region”-related genes, such as the hydrophobic-surface-binding protein A (HsbA) and cutinase-related proteins, cell wall structural proteins, melanin synthesis enzymes, chitin deacetylases, and virulence proteins being extremely enriched. These DEGs are closely related to the phenotypic responses to ET in this phytopathogenic fungus.

In filamentous fungi, cell walls are commonly darkened by accumulating DHN-melanins. These pigments are secondary metabolites produced via the polyketide synthase pathway [56]. In this study, microscopic observation of the melanized appressoria is in accordance with the gene expression data, indicating that the melanin biosynthesis activity is accelerated by ET during the spore germinating and appressorium developing stages of *C. gloeosporioides*. The roles of appressorial melanization in fungal pathogenicity has been well documented [65,66]. Melanin may contribute to the cross-linking of cell wall components in order to reinforce the rigidity of appressoria, allowing them to withstand hyper-osmosis and generate mechanical infection force [67]. Meanwhile, hyperaccumulation of melanin pigments at the appressorial surface may also contribute to fungal survival during the oxidative burst of host plants [68,69]. This study found that ET can promote the melanin-defective mutant Δ*Cgscd1* to germinate and develop appressoria as it does in the wild-type strain; however, ET had little effect on the pathogenicity of the Δ*Cgscd1* mutant. These findings suggest that melanin could be an indispensable cellular constituent being responsive to ET stimulus to determine the pathogenicity of *C. gloeosporioides*.

Another representative gene induced by ET is *CgCP1* coding for the cerato-platanin effector, reminiscent of the scenario wherein the homologous gene *Bcspl1* in *B. cinerea* was also sensitive to ET [21]. BcSpl1 is one of the most abundant proteins in the secretory group of *B. cinerea*, and the loss of this gene led to a decrease in the pathogenicity of the fungus [70]. Moreover, the cerato-platanin homologs have also been found to be required for the pathogenicity of *M. grisea* [71] and *C. gloeosporioides* [59]. This study shows that the Δ*Cgcp1* mutant is still ET-sensitive in vitro; however, exogenous ET could not promote the pathogenicity of Δ*Cgcp1*, suggesting that cerato-platanin is a vital virulence factor being stimulated by ET to accelerate host infection.

Secreted enzymes were included in the DEGs. Genes coding for chitin deacetylases (CDAs) were upregulated by ET at 4 hpi, which was phenotypically verified by differential staining of chitin and chitosan, indicating the presence of chitosan in appressoria but not in vegetative hyphae. Chitin undergoing deacetylation into chitosan in the cell wall has been associated with hyphal growth and differentiation, such as polar growth in *Aspergillus fumigatus* [72], appressorium development in *Magnaporthe oryzae* [73] and *Uromyces fabae* [74], and formation of the infection structure in *Botrytis cinerea* [54]. The transformation of chitin into chitosan in *C. gloeosporioides* thus might also be responsive to ET to accelerate appressorium development and pathogenicity. The production of chitosan plays a role in cell wall integrity and anchoring of melanins [75,76,77]. Moreover, chitin deacetylation is thought to prevent the detection of chitin as a pathogen-associated molecular pattern (PAMP) by the host immune system [78]. In the powdery mildew fungus, RNAi silencing or chemical inhibition of CDA resulted in a dramatic reduction in fungal growth, which was linked to a rapid elicitation of chitin-triggered immunity; thus, CDA represents a promising target for the control of phytopathogenic fungi [79]. However, due to the redundancy of CDA genes, the exact roles of chitin deacetylation in *C. gloeosporioides* remain open to be characterized.

Moreover, fungal extracellular cutinases can help the pathogen penetrate through the outermost cuticular barriers of host plants. Cutin monomers, produced by low levels of constitutively expressed cutinase, are prone to inducing high levels of cutinase, which can aid pathogenic fungi in penetrating the host through the cuticle, with cutin as the major structural polymer [80]. Cutinase transcription factor 1α (CTF1α) and CTF1β are involved in strongly inducing and constitutively maintaining cutinase expression, respectively [80,81]. Similarly, the HsbA is known to recruit cutinases to the hydrophobic surface [55]. The transcriptomic analysis in this study revealed that ET can stimulate the expression levels of both CTF1α, CTF1β, and HsbAs and their related cutinases during the early germinating stage of *C*. *gloeosporioides*. These findings suggest that ET signaling is capable of triggering the invasive weapons of the fungal pathogen to pierce the host.

Programmed cell death and autophagy are also known to be involved in conidial germination and appressoria differentiation in some fungal plant pathogens [28,82]. The ET-motivated appressorium formation is a series of morphogenesis involving cell cycling processes. ET treatment leads to the development of multiple appressoria from each individual spore, accompanied by cellular apoptosis as evidenced by FAD and PI staining assay. These findings imply that ET may be recognized by the fungal pathogens as a triggering signal for ripening and susceptibility in host plants, and upon the ET stimulus, the fungal spore germlings should exert all their energies until sacrificed to establish infections.

Besides ET, many other host-plant-related materials may stimulate spore germination and appressoria differentiation of pathogenic fungi, such as adhesives, cutin monomers, surface hydrophobicity, surface hardness, and topographic signals [83,84]. Several signal pathways participate in regulating the appressorium formation, such as Gα and Gβ proteins [33], the adenylyl cyclase and PKA-mediated cAMP pathway [85], calmodulin signaling [86], and the MAPK signaling pathway [37]. It is proposed that the G-protein-coupled receptor BcGPCR3 of *B. cinerea* possesses the ability to bind ET and regulate responses to ET [36], while the MAPK cascade and their protein phosphorylation are involved in regulating the ET responses of *C. gloeosporioides* [87]. This study reveals that the MAPK pathway components (*Cgmk1* and *Cgste11*) play core roles in mediating morphological and molecular responses to ET. However, the identified GPCR3 homologs (*Cggpcr3-1/2*) are responsible for partial ET responsiveness, suggesting that there are still remaining undiscovered components that could participate in ET sensing in this fungus.

Collectively, this paper reports that ET could alter transcript levels of the genes that are closely relevant to accelerating spore germination, appressoria formation, and virulence expression of *C. gloeosporioides*. Moreover, ET promoting the gene expression, morphogenesis, and pathogenicity of *C. gloeosporioides* seems to be dependent on the GPCRs and the MAPK pathway, although the upstream perception mechanisms for ET signal in the fungus still remain obscure. These findings correspond well with the hypothesis that phytopathogenic fungi could response to ET via the pathway being different from in plant [88]. Thus, it is possible to selectively block the ET perception in fungal pathogens for the purpose for disease management during fruit ripening, although achieving this goal should be based full understanding the exact ET sensing and signaling mechanisms in fungi in the future.

## Figures and Tables

**Figure 1 jof-08-00570-f001:**
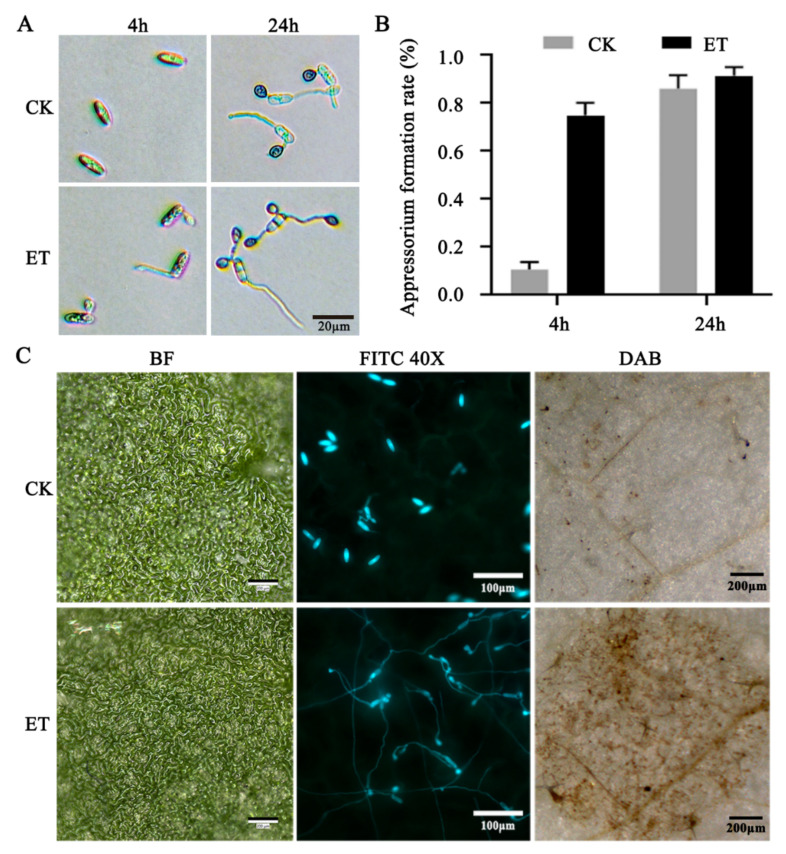
The phenotypic responses of *C. gloeosporioides* to ET during the in vitro culture and host infection processes. (**A**) Microscopic analysis of control (CK) and ET-treated samples. (**B**) The percentage of appressorium formation on glass in the CK/ET treatment groups based on calculating three replicates of 300 conidia for each sample. (**C**) Light and fluorescent microscopic visualization of *Cg**-gfp* spores germinating and infecting *Arabidopsis* leaves; the DAB panel indicates the inoculation sites after DAB staining.

**Figure 2 jof-08-00570-f002:**
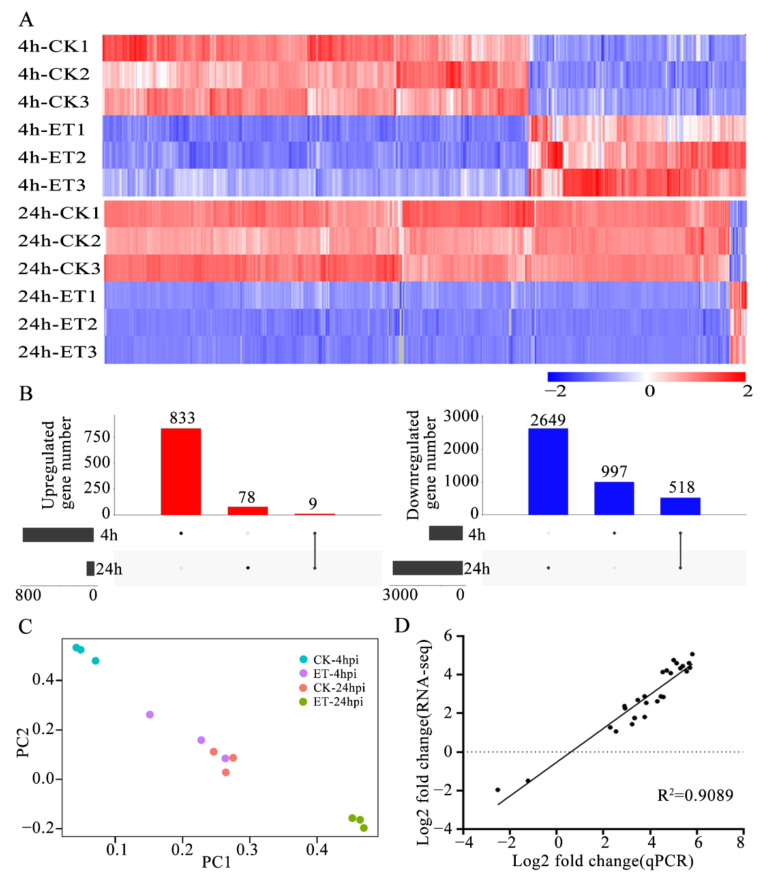
Macroscopic analysis of the transcriptome of *C. gloeosporioides* regulated by ET. (**A**) Heatmap analysis of differential genes between the CK and ET treatment groups at 4 hpi and 24 hpi. (**B**) UpSetR visualizations of intersections between the upregulated and downregulated differential genes of the CK/ET treatment groups at 4 hpi and 24 hpi. (**C**) Principal component analysis of the CK-4hpi, ET-4hpi, CK-24hpi, and ET-24hpi transcriptome data. (**D**) Correlation of gene expression values obtained by RNA-seq and qRT-PCR analysis with 27 genes, and an R^2^ value of 0.9089 was obtained by comparing the results obtained with the two techniques.

**Figure 3 jof-08-00570-f003:**
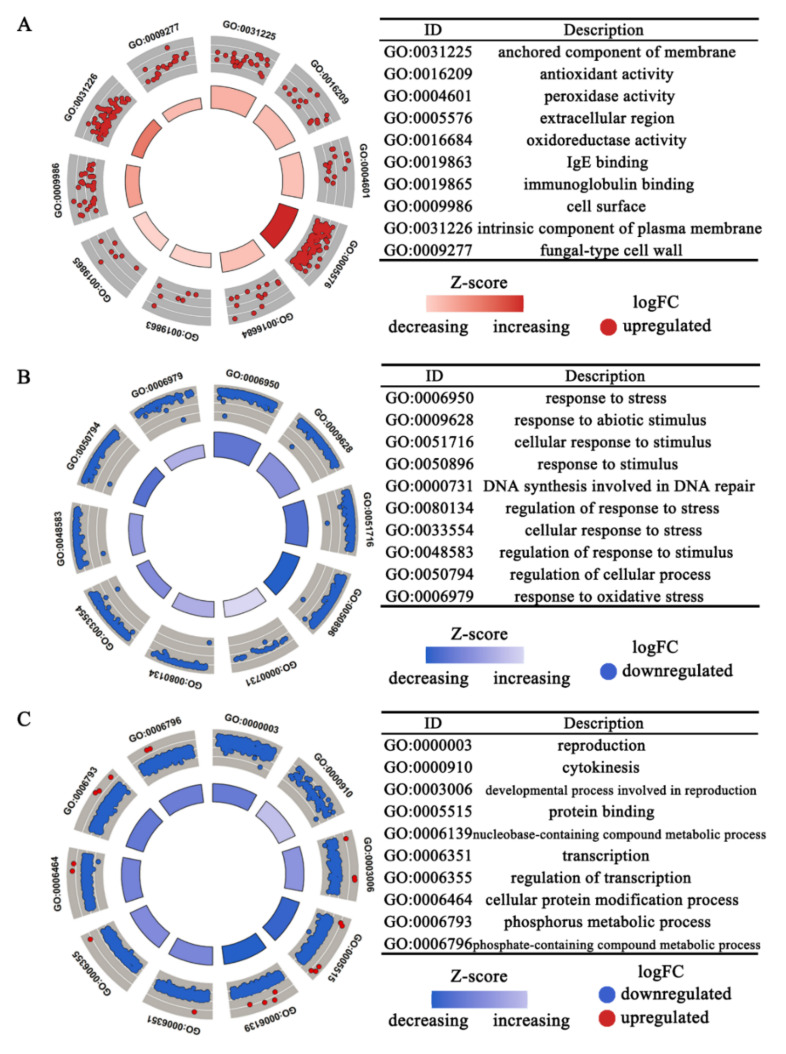
Summary of Gene Ontology (GO) terms enriched with the differentially expressed genes (DEGs) regulated by ET at 4 hpi and 24 hpi. (**A**) GOCircle showing the DEGs of the top 10 upregulated GO terms at 4 hpi. (**B**) GOCircle displaying the DEGs in the top 10 downregulated GO terms at 4 hpi. (**C**), GOCircle showing the DEGs of the top 10 GO terms at 24 hpi.

**Figure 4 jof-08-00570-f004:**
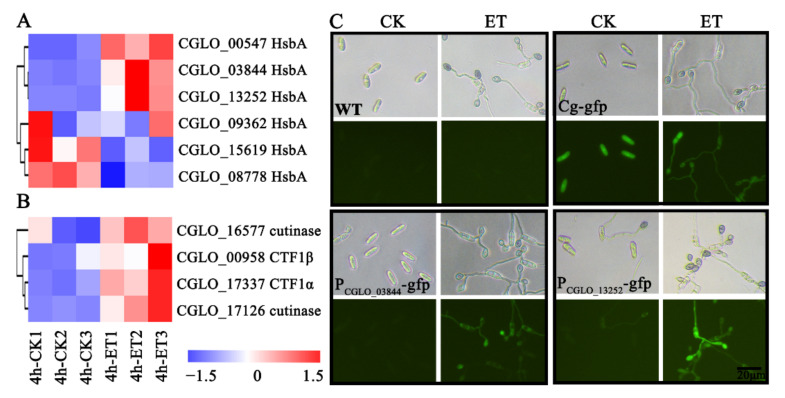
ET can significantly promote the expression of hydrophobic-surface-binding protein A genes and cutinase-related genes. (**A**) ET can significantly promote the upregulation of genes encoding for hydrophobic-surface-binding protein A (HsbA). (**B**) ET can significantly upregulate the expression levels of cutinase and its transcription factor (CTF1α and CTF1β) genes. (**C**) In contrast to the WT and *Cg-gfp* control strains, ET can induce the expression of GFP driven by the promoters of the two HsbA genes in the P_CGLO_03844_-*gfp* and P_CGLO_13252_-*gfp* transgenic strains at 24 hpi after incubation on glass slides.

**Figure 5 jof-08-00570-f005:**
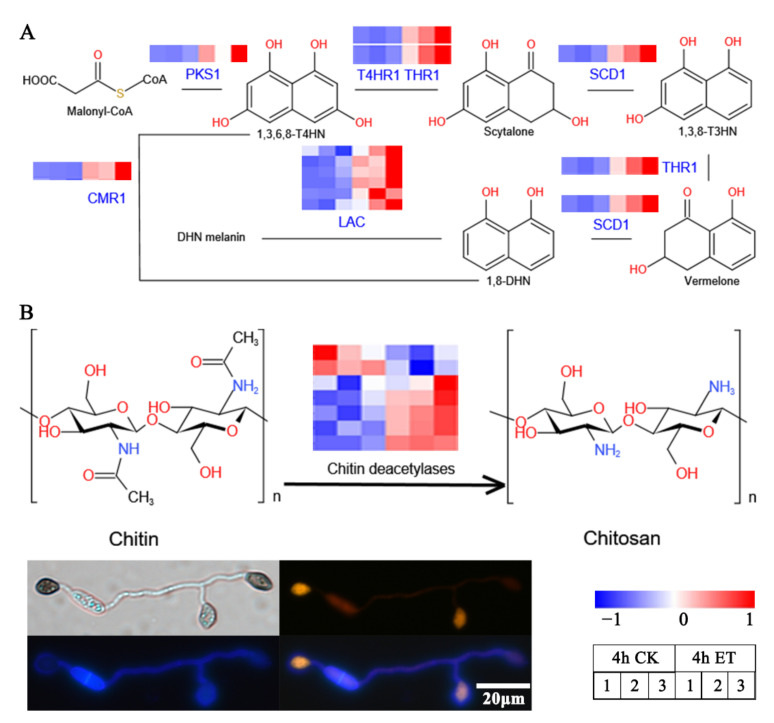
ET can significantly promote the expression of melanin synthesis genes, chitin deacetylase, and apoptosis-related genes. (**A**) Ethylene can significantly promote the upregulation of genes related to melanin synthesis. (**B**) Double-stained with Calcofluor targeting mainly chitin (blue) and Eosin Y targeting mainly chitosan (yellow). The bottom right table in the panel shows sample order information for the fungal gene expression heatmaps.

**Figure 6 jof-08-00570-f006:**
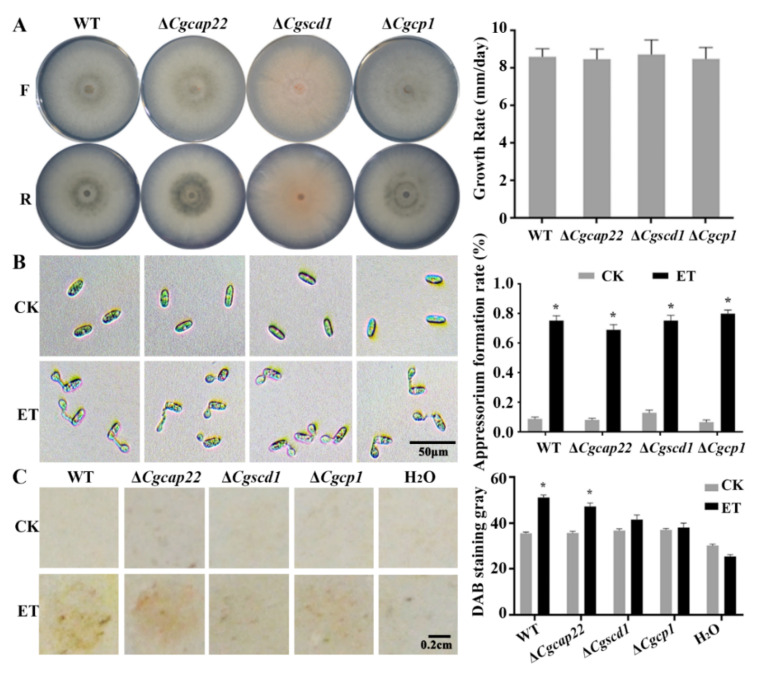
Phenotypic characterization of three mutants with disruption of the ET-responsive genes including *CgCAP22*, *CgSCD1*, and *CgCP1*. (**A**) The left two rows of photos demonstrate the front (F) and reverse (R) sides of the fungal colonies grown on PDA for 5 days; the right bar chart indicates that the colony diameters of the mutants showed no significant difference in comparison with the WT. (**B**) ET significantly promoted the appressorium formation of WT and the mutants in vitro at 4 h after incubation on glass slides. (**C**) ET treatment enhanced ROS accumulation of WT and Δ*Cgcap22* in host leaves, but caused no effect on Δ*Cgscd1* and Δ*Cgcp1* at 24 h after inoculation. For (**B**,**C**): *p* < 0.05 (*).

**Figure 7 jof-08-00570-f007:**
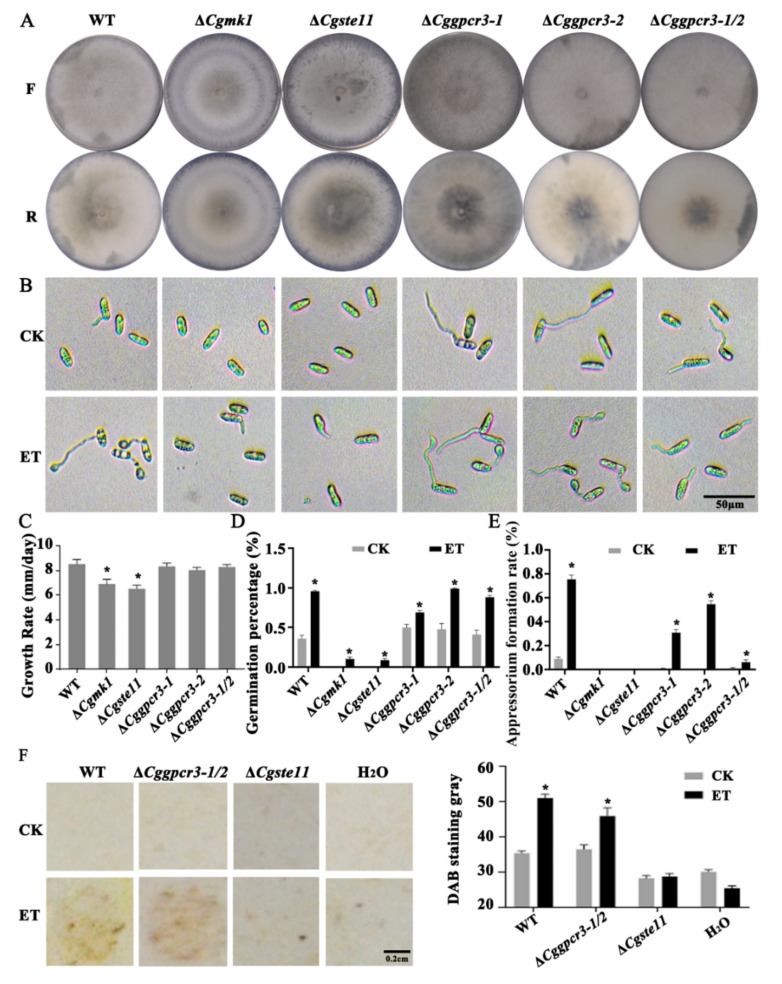
MAPK and GPCRs were involved in mediating ET signaling to promote appressorium development and pathogenicity. (**A**,**C**) The front (F) and reverse (R) sides of colony morphology and expansion rates of the strains grown on PDA for 5 days. (**B**,**D**,**E**) The effect of ET on spore germination and appressorium development at 4 h after incubation on glass slides. (**F**) The influence of ET on ROS accumulation of the plant leaves infected with WT, Δ*Cgste11*, and Δ*Cggpcr3-1/2* strains at 24 h after inoculation. For (**C**–**F**): *p* < 0.05 (*).

**Figure 8 jof-08-00570-f008:**
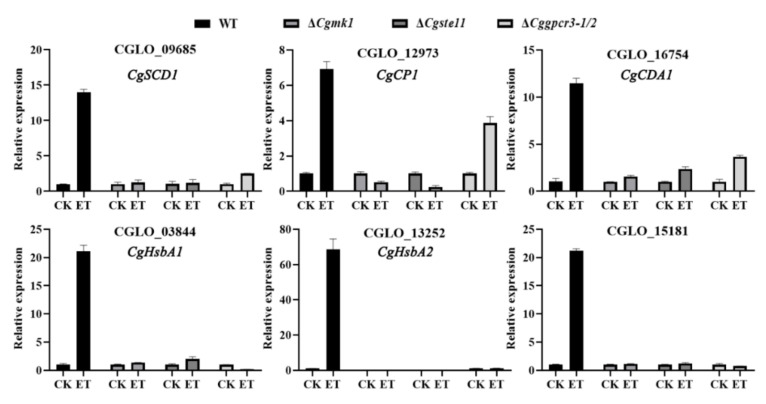
Comparative expression analysis of the ET-induced genes in the WT and Δ*Cgmk1*, Δ*Cgste11*, and Δ*Cggpcr3-1*/*2* mutants.

**Table 1 jof-08-00570-t001:** Fungal strains used in this study.

Strain Name	Genotype Description	Strain Source
WT	Wild-type *C. gloeosporioides* (EX2016-02)	[41]
*Cg-gfp*	Fluorescent strain of *C. gloeosporioides*	In this study
Δ*Cgcap22*	Appressorium structural protein null mutant	In this study
Δ*Cgscd1*	Scytalone dehydratase null mutant	[42]
Δ*Cgcp1*	Cerato-platanin protein null mutant	In this study
Δ*Cgmk1*	MAPK null mutant	In this study
Δ*Cgste11*	MEK kinase null mutant	In this study
Δ*Cggpcr3-1*	G-protein-coupled receptor 3-1 null mutant	In this study
Δ*Cggpcr3-2*	G-protein-coupled receptor 3-2 null mutant	In this study
Δ*Cggpcr3-1/2*	G-protein-coupled receptors 3-1 and 3-2 double deletion mutant	In this study
P_CGLO_03844-_*gfp*	Transgenic strain expressing GFP driven by CGLO_03844 promoter	In this study
P_CGLO_13252_-*gfp*	Transgenic strain expressing GFP driven by CGLO_13252 promoter	In this study

**Table 2 jof-08-00570-t002:** Twelve DEGs that were upregulated by ET by more than 30-fold at 4 hpi.

Id	log2FC	Description	Length (aa)	Domain
CGLO_15589	5.0674	DJ-1/PfpI family protein	248	GATase1_PfpI_2 (34–229)
CGLO_07422	4.8147	extracellular-serine-rich protein	198	Cupredoxin (53–154)
CGLO_11351	4.7586	hypothetical protein	208	SodA (5–193)
CGLO_03249	4.6034	hypothetical protein	354	M35_deuterolysin_like (168–343)
CGLO_07611	4.6008	hypothetical protein	218	LPMO_auxiliary-like (21–138)
CGLO_14431	4.5783	hypothetical protein	237	nucleoside_deaminase (65–151)
CGLO_03844	4.4491	cell wall protein	223	HsbA (29–150)
CGLO_02789	4.339	catalase	504	catalase_fungal (44–492)
CGLO_05804	4.2188	WSC-domain-containing protein	446	WSC (43–122,152–227,255–332)
CGLO_00547	4.1784	hypothetical protein	167	HsbA (4–123)
CGLO_13252	4.1765	hypothetical protein	319	HsbA (27–146)
CGLO_11882	4.018	cas1-appressorium-specific protein	241	DUF3129 (19–193)

## Data Availability

Data are contained within the article and Appendix A.

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
