# Peer review of "Ethylene Promotes Expression of the Appressorium- and Pathogenicity-Related Genes via GPCR- and MAPK-Dependent Manners in *Colletotrichum gloeosporioides"

_jof, 2022, doi:10.3390/jof8060570_

Round 1

Reviewer 1 Report

This manuscript presents well designed and operated research about the role of ethylene (ET) in the development and virulence of the plant pathogen Colletotrichum gloeosporioides. The study is of relevance because this pathogen can cause  considerable loss during fruit storage. 

The use of knock out mutations in the ET-responsive genes clearly revealed the important role of ET in the enhanced expression of appressorium development and activation of several virulence genes in the pathogen. The methods are well described and the results presented clearly and with instructive figures. The results were conclusive and discussed well in context with the state of the existing knowledge.

There are only very minor issues to be corrected.

In Figure 2, part C: the green dots had to be labeled with "ET-24hpi;

In Figure 4, part C, "ET" nedd to be corrected to "ET-4h or ET-24h; in the legneds (line 343, "C,, should be in bold and within brackets.

In Figure 5, parts A and B, the figure does not clearly show the effect of the fungal gene expression in realation to control vs. ET-treated.

In Figure 7, in part C, "groth rate" has to be corrected to "growth rate" and in the part D, "germintion percentage" has to be corrected to "germination percentage". 

Reviewer 2 Report

Review Ren et al. 2022

Ethylene regulates expression of the appressorium- and pathogenicity related genes via GPCRs and MAPK signaling components in Colletotrichum gloeosporioides.

The manuscript s by Ren et al. describes the roles of Ethylen on the differentiation of C. gloeosporioides Conidia into Appressoria.

It has been reported in 1994 that Ethylen can stimulate AP formation in certain  Colletotrichum species.

The manuscript extends on this finding and anlalysis the effects of ET on C. gloeosporioides by RNA SEQ analysis.

The original finding  could be validated, however, the effect of ET appeared to be only measurable early upon AP formation, because the neg. control can form APs in the absence of ET, albeit much more slowly.

Therefore all effects of Ethylen must be discussed in the context of timing.

As a consequence all conclusions here must be taken with a grain of salt and the conclusions and in my view the title need to be changed.

If for example genes involved in Melanin biosynthesis are induced during AP formation it does not mean that theses gene are under control of ethylen.

The same is true for any potential ethylene receptor. It has been known for a long time the MAPK pathways are required for AP formation. Reduced AP formation in such mutants is not necessarily due to a failure in Ethylen  perception.

Besides this extremely important point, the data presented are very nice and are well documented.

In summary, the manuscript describes interesting data, but the conclusion are not justified.

IF THE AUTHORS CHANGE THEIR CONCLUSIONS, THE PAPER COULD BE CONSIDERED FOR PUBLICATION AFTER CAREFUL REVISION, BECAUSE THE QUALITY OF THE ACTUAL DATA IS VERY GOOD!  

Reviewer 3 Report

The MS of Ren et al. entitled “Ethylene regulates expression of the appressorium- and pathogenicity-related genes via GPCRs and MAPK signaling components in Colletotrichum gloeosporioides” has the potential to represent a substantial contribution to the better understanding of the effects of ethylene on the initial steps of infection carried out by C. gloeosporioides.

However, essential information is missing in the material and methods section and in several parts of the MS.

Without the information in the material and methods sections it is not possible to understand how the experiments mentioned in sections 2.2 Ethylene treatment and sample preparation, 2.4 Library preparation and RNA sequencing, 2.8 Fluorescence-Based Reporter Assay, and 2.9 Appressorial formation and pathogenicity assay, were carried.

For example, in section 2.2 Ethylene treatment and sample preparation the authors mention: “Control and ET treatment were performed as shown above. After incubation for the indicated time, the supernatant liquid was gently poured out, the remaining conidia and germlings were scratched from the bottom of the dishes by using a cell scraper.”

Why are not the authors clearly indicated what concentration of ET were used (or reached – in ppm), the time of incubation, the number of replicates and so on?

Section 2.4 Library preparation and RNA sequencing. The authors indicate: “Briefly, the mRNA of three biological replicates of each condition was separated and enriched by oligo dT selection.” What was a biological replicate? What conditions? The experimental design is not presented!

In section 2.8 Fluorescence-Based Reporter Assay the authors indicate that: “The conidia of the transformants were germinated on glass slides to evaluate the effects of ET on the intensity of GFP signals as observed with a fluorescence microscope (Axio Imager 2, ZEISS, Germany).

How was the experiment performed? After how much time was the intensity GFP observed? How was the treatment with ET performed? How many times was the experiment repeated and how many replicates were used in each experiment?

In section 2.9 Appressorial formation and pathogenicity assay the authors again indicate: “Control and ET treatments were conducted as described above. After incubation for the indicated time, conidial germination and appressorium formation rates were recorded.”

Where were the treatments described (concentration of ET were used, the time of incubation, and so on)? At least, in this section, the authors mention: “For each treatment, three replicates of 300 conidia were randomly selected for statistical analysis.”

Also, “The samples were photographed every day, and the lesion areas were calculated using the ImageJ software.” How many days were the samples observed?

Other, general comments:

Please leave a space between numbers and hours 4 h not 4h, number and nanometers, 530 nm, not 530nm, Figure 7D, not Figure7D, and so on, throughout the MS.

References: italicize scientific names.

In Figure 7F use only one row of photos for the ET, similarly to panel A, colony morphology, and panel B spore germination. Indicate the time for germination and appressorium percentage.

Correct growth the rate/mm, should be growth rate (mm).

Germintion should be germination.

Appressorium percentage. Percent of what?

DAB staining gray. Why gray? How was the staining quantified?

Round 2

Reviewer 2 Report

The authors have addressed most concerns I had .

Reviewer 3 Report

The authors address well reviewer's comments.